# A Capacitive Ice-Sensor Based on Graphene Nano-Platelets Strips

**DOI:** 10.3390/s23249877

**Published:** 2023-12-17

**Authors:** Sarah Sibilia, Luca Tari, Francesco Bertocchi, Sergio Chiodini, Antonio Maffucci

**Affiliations:** 1Department of Electrical and Information Engineering, University of Cassino and Southern Lazio, 03043 Cassino, Italy; luca.tari@unicas.it (L.T.); maffucci@unicas.it (A.M.); 2E-Lectra srl, 03043 Cassino, Italy; 3Nanesa srl, 52100 Arezzo, Italy; francesco.bertocchi@nanesatech.it (F.B.); sergio.chiodini@nanesatech.it (S.C.); 4Italian National Institute for Nuclear Physics, INFN-LNF, 00044 Frascati, Italy

**Keywords:** electrical permittivity, graphene nano-platelets, ice sensors, nanomaterials

## Abstract

This paper investigates the possibility of realizing ice sensors based on the electrical response of thin strips made from pressed graphene nano-platelets. The novelty of this work resides in the use of the same graphene strips that can act as heating elements via the Joule effect, thus opening the route for a combined device able to both detect and remove ice. A planar capacitive sensor is designed and fabricated, in which the graphene strip acts as one of the armatures. The sensing principle is based on the high sensitivity of the planar capacitor to the change in electrical permittivity in the presence of ice, as shown in the experimental case study discussed here, can also be interpreted by means of a simple circuit and electromagnetic model. The properties of the sensor are analyzed, and the frequency range for its use as an ice detector has been established.

## 1. Introduction

In extreme environments, such as in aerospace applications [1] or wind turbines [2], the importance of accurate ice detection cannot be overstated. Ice sensors serve as the first line of defense against potentially catastrophic events or major damage caused by ice accretion on critical surfaces. The reliable and accurate detection of ice is also needed in standard environments, such as roads and bridges [3] or power lines [4], to ensure the desired level of safety. Several techniques for ice detection have been assessed in recent years, based on pneumatic, magnetostrictive, piezostrictive, ultrasonic, dielectric, acoustic, thermographic, microwave, and photonic sensing principles [5,6,7,8,9,10].

This large variety in ice sensor technologies provides the possibility of choosing the best solution for any given application, considering the specific requirements. However, each of the proposed technologies has advantages and drawbacks, and thus research is still needed both to optimize existing solutions and to find novel ones. Sensors based on physical contact are based on the influence of the ice on the surface acoustic waves [11], on the electrical parameters [3], on the latent energy [5], on the heat transfer on temperature sensors made by thermocouples [12], or by fiber Bragg gratings [9,13]. These sensors have the advantage of easier embedding but can lead to an underestimation or overestimation of the actual ice accumulation, being most highly sensitive to surface or near-surface contact. In addition, they are usually made of a sensing material that is different from the surrounding material, and this fact can lead to a thermal island effect, as indicated in [3] with reference to road ice detection. Methods based on indirect measurements, such as, for instance, those based on the microwave response of planar split rings [7] or on tomography imaging using ultrasonic waves [14], have been shown to be effective but usually require complex hardware and software components, thus imposing severe constraints on implementation and significant maintenance costs.

In all the techniques cited here, the sensitivity of the sensors is strongly affected by the type and thickness of ice and by environmental conditions such as temperature fluctuations and humidity levels [15,16]. Finally, it must be pointed out that, in many applications, the chosen ice sensor technology is not necessarily the best one, being selected from among those compliant with the given standards. This is the case, for instance, for ice-sensors installed in aircraft which must fulfill the strict requirements imposed by aeronautical standards such as RTCA-DO-160G [17].

For all the above reasons, the current research on ice sensors not only addresses the optimization of current technologies but also the creation of new solutions based on the use of novel materials, such as nanomaterials. Indeed, due to their specific features that may lead to unprecedented performance, these materials are definitely interesting candidates to realize novel sensing elements. Specifically, materials derived from graphene have undergone substantial research [18,19] that also suggested their use in sensing technologies [20,21,22,23,24,25], and more recently in ice-sensing applications [26].

In addition, the same materials have been proposed to realize novel thermoelectrical actuators [27,28], with several interesting implications. For instance, the use of graphene strips as heating elements opens the route to novel approaches to de-icing based on the Joule effect. This would allow the use of such techniques for aircraft applications, where they are not currently used, due to the impact of the conventional heaters, in terms of weight for instance. The authors have investigated the electrothermal properties of graphene strips which may be proposed as heating elements for de-icing systems [29,30,31]. Similar results have recently been assessed in the literature [32,33].

Starting from these considerations, the main motivation of this paper is to study the possibility of realizing an ice sensor with the same graphene strips that are used for de-icing. This possibility would open a route to the realization of an integrated system capable of sensing and removing ice.

With the scope of studying the feasibility of this idea, this paper analyzes industrial-grade graphene, which is not optimized for the specific purpose of sensing, but for that of heating. Indeed, the graphene and graphene-related materials that can be synthesized in a laboratory provide much better physical properties [34,35], but are still unfeasible for mass production due to the high costs associated with the fabrication technologies which are currently available. Alternative and less expensive materials are therefore being researched, such as nanocomposites with carbon-based reinforcements like graphene flakes, carbon nanotubes, buckyballs, and so forth. The work carried out in [36] provides a thorough analysis of such inexpensive graphene variants, comparing prices and effectiveness. It is demonstrated that materials based on Graphene Nano-Platelets (GNPs) offer a suitable trade-off between good physical qualities, mass production, and affordable costs. The GNPs are examples of industrial graphene that exhibit good properties [37] and that can be produced by using a number of industrially scalable methods, such as wet-jet milling [38], microwave irradiation [39], and liquid exfoliation [40]. The latter technique is used by Nanesa [41], who created the commercial GNP-based strips used here as an element of a capacitive sensor for ice detection.

The material production and characterization are briefly summarized in Section 2. This section also contains the design and implementation of the capacitive sensor and its modeling in terms of an electrostatic model and a simplified circuital one. The experimental characterization of the sensor performance in the presence of ice is investigated and discussed in Section 3, along with the identification of the model parameters. Conclusions are given in Section 4.

## 2. Materials and Methods

### 2.1. GNP Strips Fabrication and Characterization

The films of industrial graphene analyzed in this paper are strips with a lateral dimension of 1 cm, lengths between 10 and 18 cm, and thicknesses ranging from 55 to 75 µm. These films were created by Nanesa [41], using the following steps of a proprietary industrial process:(i)Graphene nano-platelets (GNPs) are produced through the thermal expansion and liquid exfoliation of an affordable graphitic precursor, namely intercalated expandable graphite.(ii)A blend is created by dispersing GNPs in either acetone or an aqueous solution, employing magnetic stirring, and concluding with a sonication step. If a polymeric binder is included, it is introduced during the sonication phase. In terms of mechanical properties, polyurethane (utilized in this study) or epoxy are identified as suitable binders for the objectives of this research.(iii)The mixture is then sprayed at a controlled pressure (using the semiautomatic 3-axes pantograph Computer Numeric Control plotter EXTREMA, model Basic), to realize the GNP strips.(iv)The GNP strips undergo a final treatment of calendaring (optionally followed by annealing), that compacts them and provides an optimized thickness/alignment ratio.

The size and thickness of GNPs are known to significantly influence the physical properties of the resulting composite material. For example, the impact of GNP thickness has been discussed in [42], indicating that thinner GNPs are recommended to enhance overall mechanical and thermal performance. On the other hand, the analysis in [43] focused on the effect of size/thickness aspect ratio on electrical conduction, revealing that the percolation threshold in the composite increased with higher aspect ratio values. Additionally, [44] demonstrated that the electrical conductivity of the composite improved with increasing GNP size and surface area. Consequently, to ensure stability in the behavior of GNP films, it is crucial to evaluate a fabrication process capable of controlling GNP dimensions. As shown in the Scanning Electron Microscope picture displayed in Figure 1a, the typical surface dimensions of a GNP are of the order of some tens of µm, with an average thickness of around 12 nm. Similar characteristics can be obtained by using alternative techniques suitable for industrial-scale production, such as wet-jet milling [38] and microwave irradiation [39].

Table 1 summarizes the characteristics of the industrial graphene strips analyzed in this paper (Figure 1b).

The electro-thermal properties of these GNP strips are analyzed in [29,31].

The electrical resistivity exhibited by these strips is suitable for their use as heater elements based on the Joule effect, as shown in [29]. Therefore, they are investigated with great interest in view of their potential use in industrial applications (e.g., to replace conventional ovens) and in aeronautical ones (e.g., in novel de-icing and anti-icing systems for aircraft wings). Note that a decreasing value of resistivity was observed with increasing temperatures, and thus these films qualify as Negative Temperature Coefficient (NTC) materials. This is an unusual behavior for electrical conducting materials, usually being found in non-conducting or semiconducting materials (such as silicon or germanium). As pointed out in [30], this behavior is associated with different mechanisms related to the charge transport within a single GNP flake and between two adjacent flakes. As the temperature increases, the carrier mobility within a single GNP is reduced by a shorter mean free path, but is enhanced by a higher number of conducting channels [45,46]; when the second mechanism is dominant, the temperature increase improves the electrical transport. In addition, the transport between two adjacent flakes is associated with classical contact conduction (that worsens with higher temperature) and to quantum mechanisms such as tunneling and hopping effects, which provide higher mobility at higher temperature. For instance, in [47] the electrical conductivity associated to the hopping and tunneling is reported to increase with the temperature *T* as $T^{4}$.

The relationship between resistivity and temperature is well described by the standard linear law adopted for conventional materials like copper:(1)$$\rho_{eq}(T)=\rho_{0}(1+\alpha (T-T_{0}))$$
with $T_{0}$ being the reference temperature, and α the Temperature Coefficient of the Resistance, TCR. The fitting parameters for the GNP strips and for copper are given in Table 2: compared to copper, the GNP strips exhibit a higher conductivity (more than one order of magnitude), but a negative TCR.

The thermal properties of the GNP material analyzed in this paper are instead investigated in [31]; Table 3 summarizes the estimated emissivity (ε) and thermal conductivity (*k*). As expected, the thermal conductivity value of this industrial graphene is lower than the values expected for pure graphene (that can rise by up to thousands of W/mK, e.g., [48]), but is still of interest, being comparable to those exhibited by conducting materials usually adopted for industrial thermal and electro-thermal applications (like copper, as reported in Table 3). In [29], an example of the use of these GNP strips as heater elements based on the Joule effect is provided, exploiting their good electrical and thermal properties.

### 2.2. ICE Sensor Operating Principle and Design

Keeping in mind the possibility of realizing heaters based on the GNP strips previously introduced, in the following, we investigate the idea of using the same strips to also realize an ice sensor, so that a potential device can be envisaged that is able to detect the ice and remove it.

Therefore, following the geometry suggested by the strips, the ice sensor here is designed as a planar capacitor with the upper face exposed to air, water, or ice, whose capacitance values vary as a consequence of the huge difference between their relative permittivity (dielectric constant), as reported in Figure 2 [49]. The relative permittivity of water and ice is significantly higher than air, and depends, in a different way, on the frequency at which the capacitor is working.

The idea developed here is of using each GNP strip as the central arm of a planar capacitor, whose cross section is depicted in Figure 3 (“Graphene strip”). The external arm is provided by a conventional conductor, here chosen in copper (“External arm”, in orange). A guard ring of conventional conductor is also designed (“Guard ring”, in orange), that provides an additional degree of freedom, since it can be connected either to arm 1 or arm 2, thus changing the topology of the capacitor. The top surface is in contact with air or ice, depending on the operating conditions.

The design in Figure 3 has been implemented in the multi-layered printed circuit board (PCB) in Figure 4, where the horizontal tracts of the conductors 2 and 3 are realized by metallization in different layers, and vertical tracts are made by using vias. The green regions are instead made by FR4 dielectric. The PCB, with surface dimensions of 108 × 15 mm, has been designed for enabling the 4-probe techniques, with two amperometric cables (in orange and brown in Figure 4) separated by the two voltmetric ones (in black and red in Figure 4).

### 2.3. Electromagnetic and Circuital Models

In order to theoretically investigate the behavior of the sensor, the circuit model depicted in Figure 5 was proposed. In this circuit, the GNP arm is modelled as a parallel R-C element, taking into account the high sensitivity of the complex permittivity of graphene to environmental conditions. Here, the resistance $R_{GNP}$ also includes the contact term. The other arm, made of copper, is instead modelled only by the equivalent resistance of the arm itself, $R_{Cu}$, that also includes any contact term. The core of the sensor is given by two parallel capacitances: $C_{int}$, which is associated with the inner part inside the PCB (FR4 dielectric), and $C_{ext}$, which is related to the displacement filed lines developing outside the capacitor. The element that is supposed to be mostly affected by the presence of air/ice is $C_{ext}$, but an effect can be also observed $R_{GNP}$ as a consequence of the varying temperature values, see (1).

The estimation of the bulk capacitance $C_{b}=C_{int}$ + $C_{ext}$ in presence and absence of ice has been carried out by using a simple 2D model developed in COMSOL Multiphysics (Ver. 6.2), as shown in Figure 6. As detailed in the Results section, the best option for the sensor sensitivity was found to be that of putting the guard to the same potential as the central graphene arm. Therefore, the sensor cross section described in the previous section has been further simplified by considering the graphene strip and the guard ring as a single conductor made of copper, with $\varepsilon_{r}=1$ (show in Figure 6). In this way, this model does not contain the capacitance $C_{GNP}.$ The second arm is also made of conventional conductor with $\varepsilon_{r}=1$, while the dielectric parts in the FR4 substrate have been assigned a value of $\varepsilon_{r}=4.5$. The element “Air/Ice” in Figure 6 is the area upon the sensor, and it represent both air ($\varepsilon_{r}=1$) or ice ($\varepsilon_{r}=80$) according to the operational conditions considered. Finally, between area 4 and the sensor itself, a thin “Plastic layer” (25 µm thickness) has been interposed (with $\varepsilon_{r}=3.5$) to adhere to the real conditions in which the experiments have been carried out. Indeed, this plastic was needed to avoid the GNP strip being in direct contact with the ice.

The results of the solution to the electrostatic problem are displayed in Figure 7, which shows the distribution of the electrical field (black arrows) and the electrical charge surface density (colored map). As shown in Figure 7a, the field (and the charge density) are confined within the PCB structure when it is surrounded by the air. When the air is replaced by ice (Figure 7b), the steep increase in the dielectric constant is responsible for the field spreading externally through the PCB (fringe fields), hence modifying the capacitance value. The electrical capacitance is then extracted from the solution to the electrostatic problem.

### 2.4. Experimental Setup

This section discusses the methodologies and the measurement process adopted in the experimental characterization of the ice sensor. The capacitance of the sensor has been evaluated by measuring its electrical impedance with the set-up conceptually described in Figure 8: the sensor is connected to a *GW-INSTEK LCR-8101G* (Gw Instek: Taipei, Taiwan) impedance analyzer, and a PC is used to drive the instrument via an RS-232C interface (sanwa supply: Okayama, Japan). Regarding the measurement process adopted, Impedance Spectroscopy has been used to investigate the response of the sensor to changes in frequency. In detail, the 20 Hz–1 MHz range was explored, sampling three points per decade in logarithmic steps (e.g., 1-2-5-10...). As shown in the set-up diagram, the connection between the instrument and the sensor has been realized by using a 4-probe configuration, consistent with the features of the PCB sensor (Figure 4). The sensor can possibly be inserted in a climatic chamber (ACS DY110, Angelantoni: Massa Martana (PG), Italy) in order to control temperature and humidity.

The impedance was measured between the external arm of the planar capacitor (Figure 3) and the graphene strip was set at the same potential as the guard ring. This configuration was the best one in terms of sensitivity, being associated with the smallest gap between the arms (equal to 0.1 mm). In line with the sensor model adopted in Figure 5, an R-C parallel model has been chosen for the impedance analyzer to directly detect the capacitance value.

Prior to the measurement process, a calibration of the impedance analyzer was performed to exclude any contributions from the parasitic, capacitive, or inductive effects of the set-up. This phase is fundamental for this type of instrument and must be performed each time the instrument is used in a new environment, or if the test set is changed. The calibration consists of two phases, the so-called *open circuit* and *short circuit* ones: in the first, the instrument clips are spaced an equal distance apart from the normal test position; in the second, the instrument clips are connected by a short circuit.

In the measurement campaign performed, the operation of the sensor under different working and environmental conditions has been characterized. In detail: (i) at fixed or varying temperature and humidity; and (ii) in the absence or in the presence of different types of ice. To this end, to assess sensitivity to the environment, the sensor has been characterized inside the cited climatic chamber as capable of controlling the temperature and the percentage of humidity in the environment. The temperature values considered are: −40, −20, 0, 20 °C. While the humidity values selected are: 0, 10, 25, 50%.

Finally, three types of ice samples (clean ice) are considered, differing in their dimensions, as listed in Table 4. The ice thickness values have been suggested for the specific use of the sensor in aerospace applications, where the maximum ice thickness is limited to 30 mm. Specifically, the sensor should be able to provide an alert signal to the control unit at the beginning of the ice formation during flight (few mm thickness). Instead, the values of the ice width are chosen to investigate different cases of the coverage of the planar capacitance: minimum (10 mm), medium (12 mm), and very large (28 mm).

In order to realize a metrologically robust characterization of the sensor, a stability analysis was performed. Specifically, in order to assess and estimate the ability of the tested sensor to respond more or less consistently to the same stimulus, 30 measurement tests were conducted for each considered working condition. In this way, it was possible to obtain an indication of sensor stability and repeatability.

In particular, since the causes of possible variations in terms of response can be many and varied (instrumentation, environmental perturbations, real variations in the measurand, etc.), the tests were not only repeated for each operating condition in the presence and absence of ice but also considering variations in environmental conditions. As shown in Section 3, for each test, the response at each frequency was given in terms of the average value over the repeated measurements and the corresponding standard deviation, which is therefore representative of the stability of the sensor.

## 3. Results and Discussion

### 3.1. Experimental Results

The results obtained from the characterization campaign are shown below, in terms of the mean and standard deviation (σ) of the sensor capacitance under the different conditions analyzed. Firstly, the sensor response in the absence or presence of ice is analyzed to validate its sensitivity and selectivity in a wide frequency range from tens of Hz to 1 MHz. Next, the stability of the sensor response under varying environmental conditions is investigated by changing temperature and humidity values. Finally, the possible working ranges of the sensor as an ice detector are studied for three different coverage factors (K·σ). The coverage factor K is related to the probability, also called the confidence level, that the measured value belongs to a certain interval [50]. In the reasonable assumption that the random effects affecting the measurements have a normal distribution, the probability that the true value of the measurand is within the confidence interval is approximately 68.4% for K = 1, 95.4% for K = 2 and 99.7% for K = 3.

Figure 9 shows the behavior of the capacitance offered by the sensor as the frequency varies in the absence and presence of the three different types of ice considered. The types of ice, in relation to Table 4, are indicated by “A”, “B”, and “C”. For air, the same nomenclature is used to indicate the correlation between the tests in the presence and absence of ice performed on the same test day. The tests have been carried out in the climatic cell and refer to a working temperature of −20 °C. The solid (dashed) lines refer to the presence (absence) of ice. As pointed out for each condition, 30 measurement tests have been carried out: the lines with markers represent the average value of the measured capacitance, whereas the lines without markers are associated with the standard deviation.

The figure clearly shows the ability of the sensor to discriminate between the presence or absence of ice for frequency values up to about 300 Hz (detector operation). The sensitivity is much more pronounced at lower frequency values (tens of Hz), because, in that range, the difference between the dielectric constants of ice and air is more pronounced (see Figure 2), hence the effect of the fringe fields shown in Figure 7b is higher.

The behavior of the permittivity shown in Figure 2 is consistent. In terms of selectively, in the same low frequency range where the sensor is highly sensitive, it also shows good selectivity, being able to discriminate among the three types of ice considered. The sensitivity and selectivity tend to disappear for frequencies higher than 300 Hz.

As for the baseline response in the absence of ice, it is possible to note an almost stable behavior in the capacitance offered by the sensor as the frequency varies, except for a small variation in the output approximately in the range (10^3^–10^4^) Hz. This aspect is probably related to the presence of a slight resonance in the system at those frequencies. Finally, for frequency values below 300 Hz, the capacitance values assumed by the sensor in the absence of ice fall in the range (205–230) pF.

Similarly, for the sake of completeness, Figure 10 and Figure 11 show the trends obtained considering a coverage factor of K = 2 and K = 3, respectively. The considerations are the same as in the previous figure, except that, as the coverage factor increases, there is a reduction in the frequency range in which the sensor behaves as a detector. The range of the capacitance values spanned by the sensor in the absence of ice is more or less the same.

To assess the stability of the sensor to environmental changes, the baseline response in the absence of ice has been studied by varying temperature and humidity. Figure 12 shows the results obtained at varying temperature values for a fixed humidity of 0%. The dotted lines represent the curves in the absence of ice at −20 °C, previously shown in Figure 9. In the operational range of the sensor (below 300 Hz), the baseline capacitance falls in the range (195–220) pF, thus having a fluctuation comparable to that observed for the repeated tests (black curves).

Similar considerations apply when studying the stability of the sensor output under different humidity conditions (fixed temperature at 20 °C) in the absence of ice. The results are reported in Figure 13, which also contains the curves obtained in the absence of ice previously shown in Figure 9 (black dotted lines). In general, the stability of the sensor in the working range identified by the detector is evident.

As a final analysis, Figure 14, Figure 15 and Figure 16 show, for each coverage factor *K* considered, the equivalent graph representative of the range where the sensor works as an ice-detector, obtained by using the average trends and maximum extremes *± K·σ* reported in Figure 11, Figure 12 and Figure 13. Specifically, the intervals reported here represent the possible ranges of capacitance values within which the sensor can respond with a certain level of confidence related to the coverage factor. In particular, we have, for the factors *K =* 1, 2, and 3, respectively, a confidence level of: 68.4, 95.4, and 99.7%. Once the desired confidence level is chosen, the frequency interval where the sensor works as an ice detector is identified where there is no overlap between the curves in the presence and absence of ice.

Furthermore, for the sake of completeness, a summary of Figure 14, Figure 15 and Figure 16 is given in Table 5. The table entries “1” and “0” indicate the condition when detector operation is permitted or not, respectively. As pointed out, this result strongly depends on the adopted coverage factor *K* (that is, on target confidence level). In particular, in the extreme case *K =* 3, with the extremely high confidence level of 99.7%, there is operation is detected only in the frequency range (20 ÷ 50) Hz.

### 3.2. Model Interpretation of the Results

The experimental results can be interpreted by means of the simple circuit model in Figure 5. Given the values of resistivity for copper and GNP strips (see Table 1), $R_{Cu}$ is negligible with respect to $R_{GNP}$, and therefore the model in Figure 5 can be simplified in the series $R_{GNP}$ − $C_{b}$. By denoting, with $R_{m}$ and $C_{m}$, the values measured by the impedance analyzer with its R-C parallel model, and by applying the correspondence between series and parallel models, the model and measured parameters are related as follows:(2)$$R_{GNP}=\frac{R_{m}}{1+\omega^{2}C_{m}^{2}R_{m}^{2}};\;C_{b}=C_{m}\left(1+\frac{1}{\omega^{2}C_{m}^{2}R_{m}^{2}}\right).$$

An estimation of the static value of the bulk capacitance $C_{b}$ is obtained by using the simple electrostatic model introduced in Section 2.3, which provides the values of capacitance reported in Table 6 for the two extreme cases of air and ice, and which are in good agreement with the low frequency values of capacitance reported in Figure 9. To investigate the AC response, a 1-order relaxation model can be adopted for the permittivity of the ice:(3)$$\varepsilon_{ICE}=\varepsilon_{1}+\frac{\varepsilon_{2}}{1+\omega \tau }$$
where $\varepsilon_{1}$ and $\varepsilon_{2}$ are the values associated to the DC and to the high frequency asymptote, respectively. The relaxation time τ appearing in (3) can be directly estimated by using (2) as $\tau =R_{m}C_{m}$, and is clearly related to the presence of the graphene strip and, in particular, to its resistance.

By using the parameter values obtained for the case-study analyzed here, the circuit model provides the sensor response plotted in Figure 17, which is in very good agreement with the experimental results.

## 4. Conclusions

In this work, the potential use of a graphene-based planar capacitor as an ice detector has been successfully proven. A printed circuit board which hosts the graphene-nanoplatelets strip, realizing the planar capacitor, has been designed and developed, whose electrical impedance is measured in terms of the presence or absence of ice. The same design can be used to feed the graphene strip with an electrical current high enough to produce heat via the Joule effect; hence, an integrated system could be envisaged that uses the same element (a graphene strip) to detect and remove the ice.

The experimental results highlight the possibility of using the sensor as an ice detector based on the variation in its electrical capacitance in a low frequency range (up to 300 Hz), where the difference in terms of relative dielectric constant between air and ice is higher (1 to about 80). The repeatability and stability of the sensor’s response to environmental changes are analyzed, assessing a relationship between the working frequency range and the desired level of confidence. The detector is shown to work up to 200 Hz with a confidence level of 95.4%, and up to 50 Hz with a confidence level of 99.7%.

The results can be interpreted by using a circuit model that considers the effect of the internal (inside the PCB) and external capacitances, as well as the electrical resistance of the graphene strip. Despite its simplicity, the model is able to reproduce the experimental results with a good agreement.

Future work will address the sensitivity of the sensors to ice thickness and to different types of ice.

## Figures and Tables

**Figure 1 sensors-23-09877-f001:**
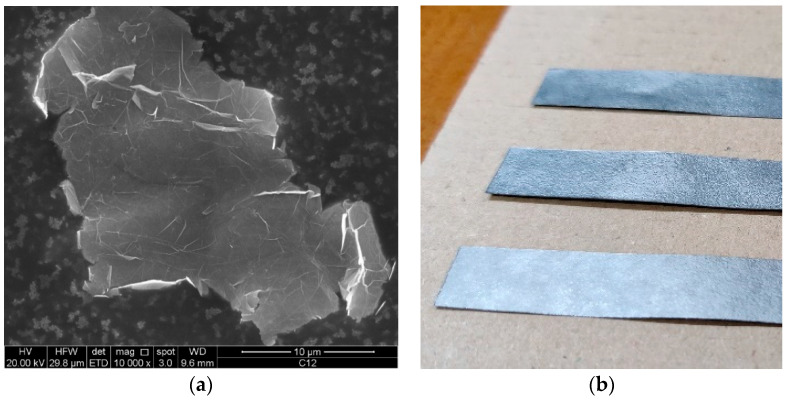
A single GNP in a Scanning Electron Microscope picture (**a**); macroscopic strips (**b**).

**Figure 2 sensors-23-09877-f002:**
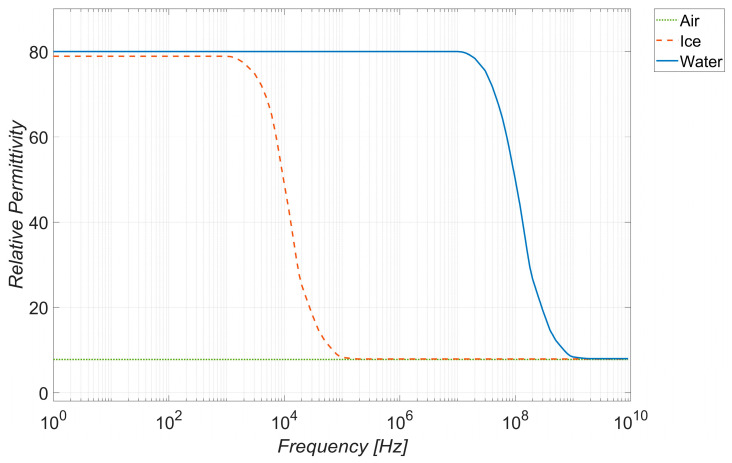
Relative permittivity of water and ice compared to air, vs. frequency, according to the data provided in [49].

**Figure 3 sensors-23-09877-f003:**
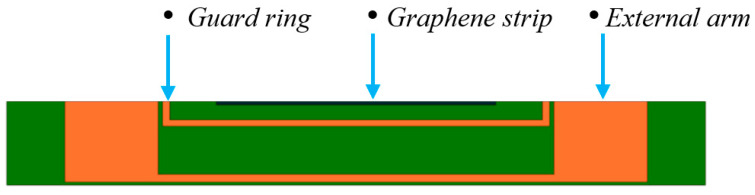
Cross-section of the designed ice sensor (planar capacitor).

**Figure 4 sensors-23-09877-f004:**
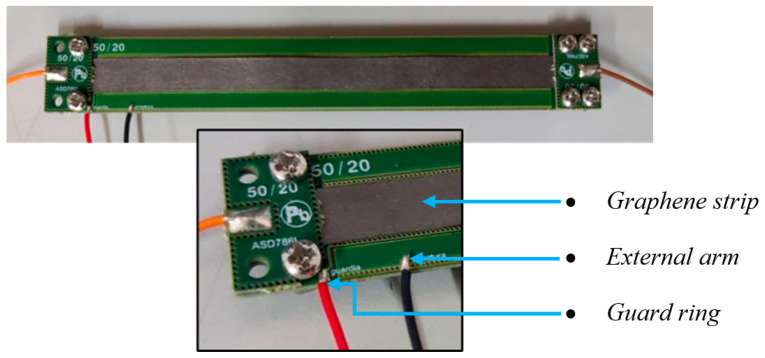
Realization of the ice sensor with a PCB structure. The central gray strip is the graphene element. The amperometric cables (orange and brown cables) are separated from the voltmetric ones (black and red) to enable the 4-probe measurement technique.

**Figure 5 sensors-23-09877-f005:**
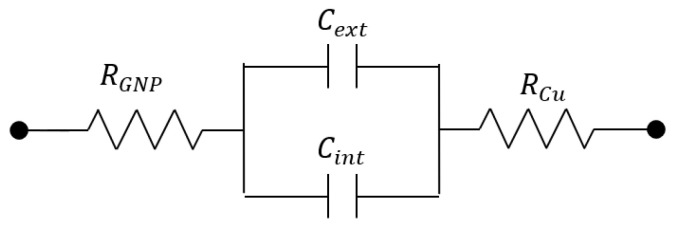
Equivalent electrical circuit for the proposed ice sensor.

**Figure 6 sensors-23-09877-f006:**
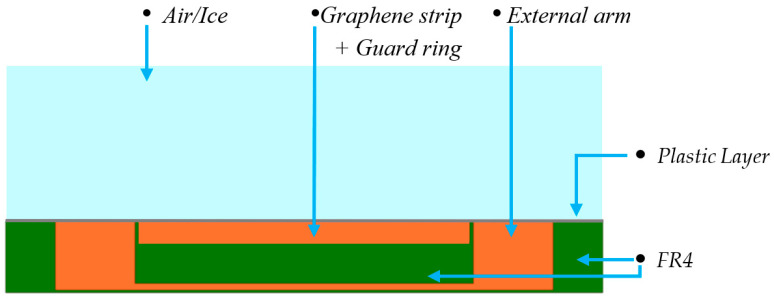
The 2D simplified model implemented in COMSOL to extract the bulk electrostatic capacitance.

**Figure 7 sensors-23-09877-f007:**
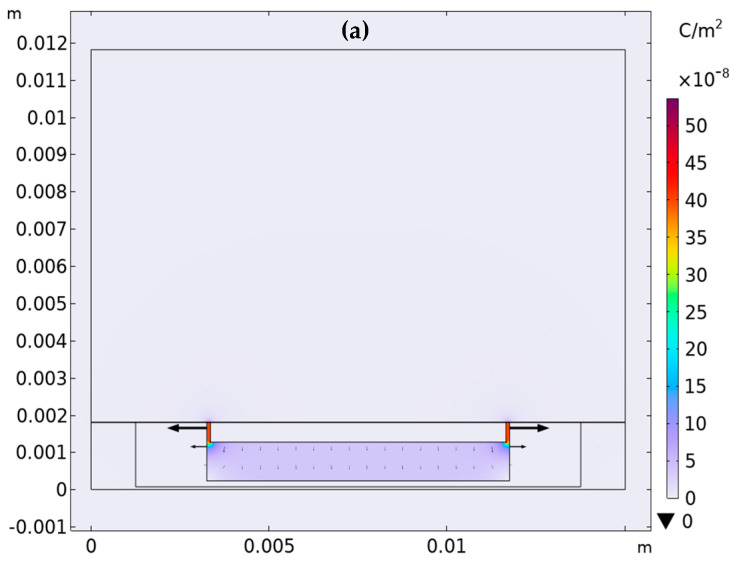
Distribution of the electric displacement field (arrows) and of the electrical charge surface density (colored map) in the presence of air (**a**) and ice (**b**).

**Figure 8 sensors-23-09877-f008:**
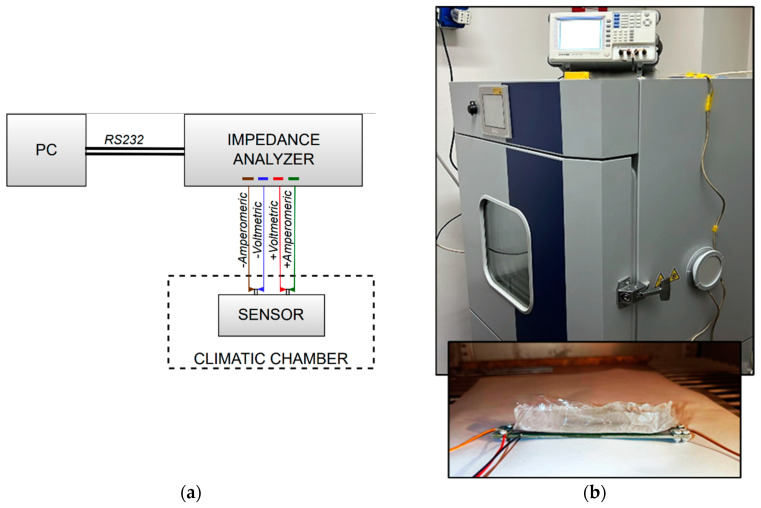
The measurement set-up, where the electrical impedance of the sensor (placed inside a climatic chamber) is measured by an impedance analyzer, supervised by a PC via an RS232C interface: (**a**) schematic; (**b**) picture of the whole setup (**top**), with the detail of the sensor covered by ice (**bottom**).

**Figure 9 sensors-23-09877-f009:**
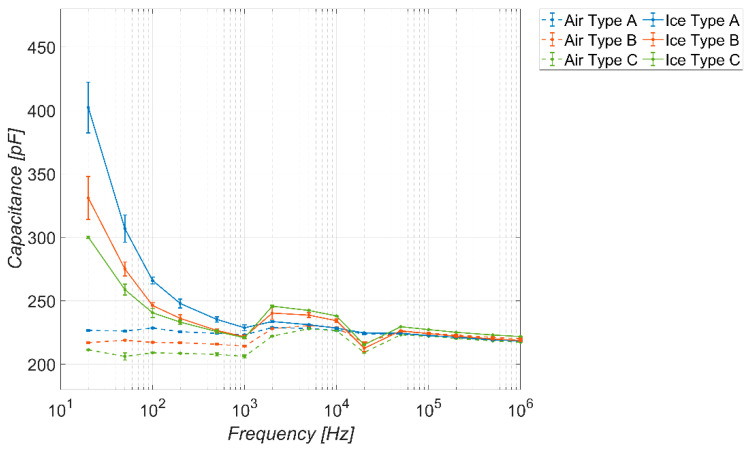
Measured capacitance of the sensor versus frequency, in the absence and presence of different types of ice. Temperature −20 °C, coverage factor *K* = 1.

**Figure 10 sensors-23-09877-f010:**
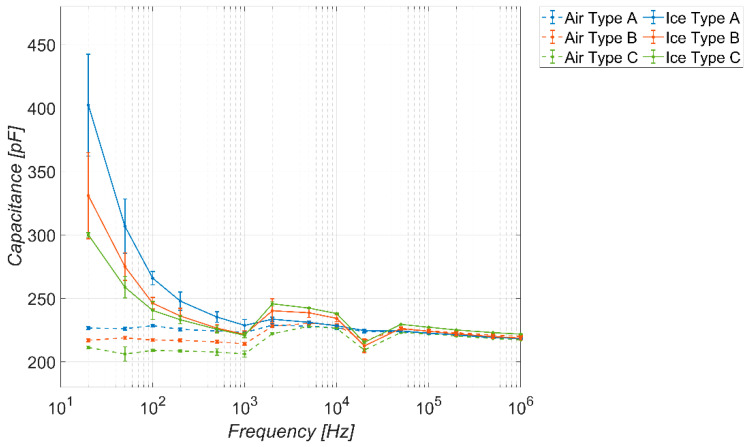
Measured capacitance of the sensor versus frequency, in the absence and presence of different types of ice. Temperature −20 °C, coverage factor *K* = 2.

**Figure 11 sensors-23-09877-f011:**
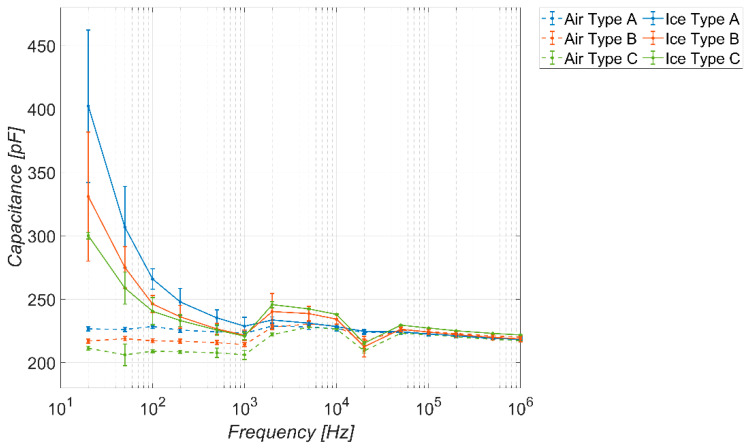
Measured capacitance of the sensor versus frequency, in the absence and presence of different types of ice. Temperature −20 °C, coverage factor *K* = 3.

**Figure 12 sensors-23-09877-f012:**
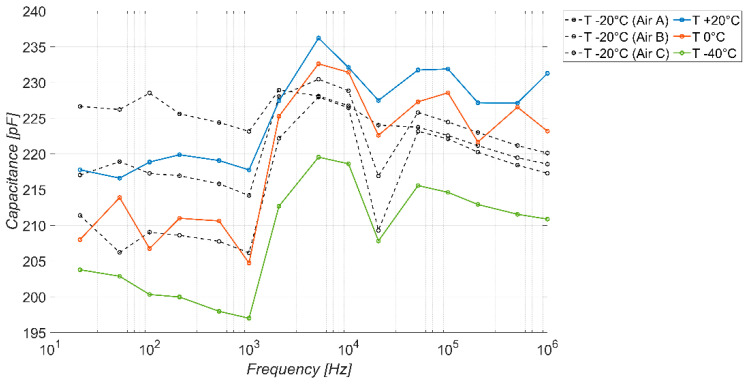
Measured capacitance (average value) of the sensor versus frequency, in the absence of ice, with varying temperature values and 0% humidity.

**Figure 13 sensors-23-09877-f013:**
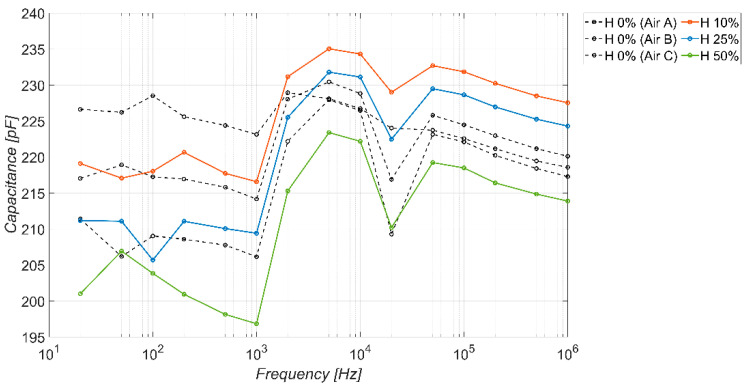
Measured capacitance (average value) of the sensor versus frequency, in the absence of ice, with varying humidity values, at 20 °C.

**Figure 14 sensors-23-09877-f014:**
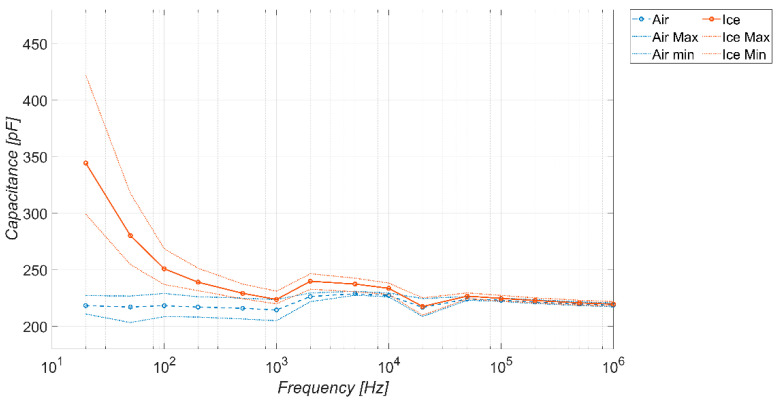
Sensor response in the presence and absence of ice with a coverage factor *K* = 1, corresponding to a confidence level of 68.4%.

**Figure 15 sensors-23-09877-f015:**
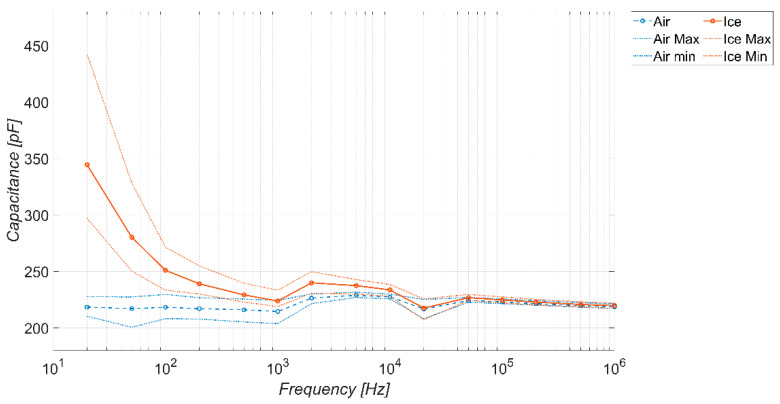
Sensor response in the presence and absence of ice with a coverage factor *K* = 2, corresponding to a confidence level of 95.4%.

**Figure 16 sensors-23-09877-f016:**
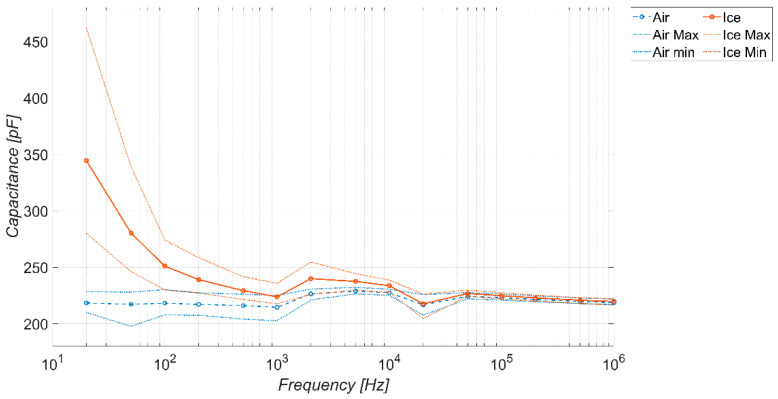
Sensor response in the presence and absence of ice with a coverage factor *K* = 3, corresponding to a confidence level of 99.7%.

**Figure 17 sensors-23-09877-f017:**
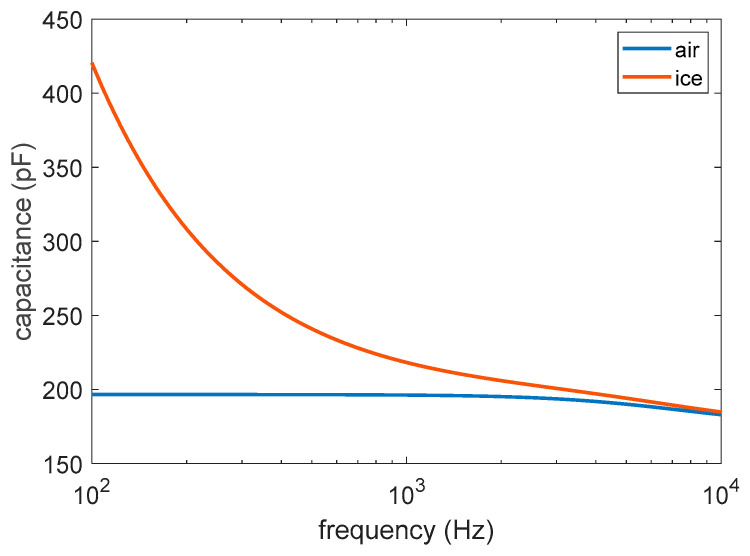
Capacitance values estimated by means of the equivalent model in Figure 5.

**Table 1 sensors-23-09877-t001:** Characteristics of the graphene strips analyzed here: G-PREG (95/5) by Nanesa [41].

Material	%GNPs	Binder	Thickness (µm)	Length (cm)	Width (mm)
G-PREG (95/5)	95	Polyurethane 5%	75	10	6

**Table 2 sensors-23-09877-t002:** Parameters for the resistivity model in (1), T_0_ = 20 °C: GNP analyzed here vs. copper.

Material	$\rho_{0}$ (µΩm)	α (1/°C)
Cu	1.68	3.90 × 10^−3^
G-Preg (95/5)	15.58	−1.37 × 10^−3^

**Table 3 sensors-23-09877-t003:** Thermal emissivity (*ε*) and conductivity (*k*) at 20 °C: GNP analyzed here vs. copper.

Material	*ε*	k (W/mK)
Cu	0.65–0.88	386–395
G-Preg (95/5)	0.53	295.5

**Table 4 sensors-23-09877-t004:** Different dimensions of the considered ice samples.

Ice Sample	Thickness (mm)	Width (mm)
Type A	3	12
Type B	12	28
Type C	15	10

**Table 5 sensors-23-09877-t005:** Summary of operating conditions of the sensor as an ice detector (value 1) at various *K*.

K	20 Hz	50 Hz	100 Hz	200 Hz	500 Hz	1 kHz	2 kHz	5 kHz	10 kHz	20 kHz	50 kHz	100 kHz	200 kHz	500 kHz	1 MHz
**1**	1	1	1	1	0	0	1	0	0	0	0	0	0	0	0
**2**	1	1	1	1	0	0	1	0	0	0	0	0	0	0	0
**3**	1	1	0	0	0	0	0	0	0	0	0	0	0	0	0

**Table 6 sensors-23-09877-t006:** Numerical estimation of the bulk capacitance of the sensor as modelled in Figure 6, in the presence of air (*ε* = 1) or ice (*ε* = 80).

Capacitance	*ε* = 1	*ε* = 80
*C_b_* (pF)	178.5	392.0

## Data Availability

The data presented in this study are available on request from the corresponding author. The data are not publicly available due to their use in ongoing projects.

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
