# Peer review of "A Capacitive Ice-Sensor Based on Graphene Nano-Platelets Strips"

_sensors, 2023, doi:10.3390/s23249877_

Round 1
Reviewer 1 Report
Comments and Suggestions for Authors
Title: A capacitive ice-sensor based on graphene nano-platelets strips
Indeed, the manuscript is well-written and easy to follow. There are some points that need to be clearly known.
- Please include a picture of the experimental setup (impedance analyzer, sensor etc.) along with the Schematic of the measurement set-up (figure 8).
-The novelty of the work should be clearly highlighted (in the abstract as well as in the conclusions).
-It is better to list a comparison table to compare results with previous work.
Comments on the Quality of English LanguageMinor editing of English language required.
Author Response
REV: 1 Indeed, the manuscript is well-written and easy to follow. There are some points that need to be clearly known.
>AU: We would like to thank the reviewer for this comment and for the valuable suggestions.
REV: 2 Please include a picture of the experimental setup (impedance analyzer, sensor etc.) along with the Schematic of the measurement set-up (figure 8).
>AU: Following this suggestion, in Figure 8 we have added a photo of the experimental setup, along with its schematic.
REV: 3 The novelty of the work should be clearly highlighted (in the abstract as well as in the conclusions).
>AU: We have rewritten a large part of the introduction to better highlight the novelty, with a more detailed analysis of the state or the art and a better definition of the scope of the paper (see sentences highlighted in yellow). As requested by the reviewer, the following sentences have been added to the abstract and to the conclusions, too. In the abstract;
“The novelty of this work resides in the use of the same graphene strips that can act as heating elements via the Joule effect, thus opening the route for a combined device able to both detect and remove ice.”
In the conclusions:
“The same design can be used to feed the graphene strip with an electrical current high enough to produce heat via the Joule effect; hence, an integrated system could be envisaged that use the same element (a graphene strip) to detect and remove the ice.”
REV: 3 It is better to list a comparison table to compare results with previous work.
>AU: previous work on these graphene strips has been addressed to the electro-thermal characterization, and the comparison is given in Tables 2 and 3 (the latter has been enriched with the values related to copper). We did not have previous results about the use of this material as element of ice sensors, since this is our first work on this topic. In addition, the following sentence has been added to comment the comparison results shown in Table 3:
“As expected, the thermal conductivity value of this industrial graphene is lower than the values expected for pure graphene (that can rise up to thousands of W/mK, e.g. [48]), but still of interest, being comparable to those exhibited by conducting materials usually adopted for industrial thermal and electro-thermal applications (like copper, as reported in Table 3).”
Reviewer 2 Report
Comments and Suggestions for Authors
This paper explored the possibility of using GNP strips for ice detection and removal. I listed a couple of comments below to address -
1. Figure 12 to Figure 16 are mislabelled as Figure 22 to 66.
2. Figure 9 to 16 have very poor resolution. Please fix them.
3. Line 41 to 50 - references are needed to justify the claims that existing ice sensors "may not capture the full extent..."; "may vary depending on the type..."; "temperature fluctuations..."; "forms of ice..."; "cost". All these are very important metrics to evaluate certain sensors. Therefore, a state-of-the-art literature search has to be comprehensive, such that readers can appreciate why the proposed graphene sensor is good.
4. Line 63 - typo - "Authors" should be "authors". There should not be a comma before "that".
5. Line 76 - typo - "The work" should be "This work".
6. Line 82 - typo with an "o" - it seems to be "and"
7. Line 118 to 120 - how about an alternative method? Can it achieve similar metrics? Please comment and provide references. Also, does the mentioned parameters (30 um particle, etc) have anything to do with the SEM image in Figure 1? If so, what is the correlation? It is better to be more illustrative.
8. Line 136 - What is the physical reason for the negative temperature coefficient? Why it is unusual? Silicon's mobility (lightly doped) also has a negative dependence on temperature. Please comment and provide references for graphene.
9. Line 137 - why NTC materials are of great interest for sensing? Please justify.
10. Table 3 - it is advisable to put some other materials's k for comparison such that the benefit of GNP can be shown.
11. Figure 2 is NOT your original data. I suggest removing it and only citing the values from Ref 40. Otherwise, you should get permission from Ref 40 to place their data in your paper.
12. Figure 3 - in the caption, the area 1/2/3 should be specified with words.
13. Figure 4 - in the zoom-in picture, a couple of labeling of which is which corresponds to Figure 3 would be helpful.
14. Figure 6 - what are 1,2,3,4,5? Please specify them in your caption so that readers can understand them more easily.
15. Line 229-232 - No need to show equations (2)-(3) as they are textbook context, nothing new.
16. Figure 7 is very poorly presented - What am I looking at? What is the scale bar of the magnitude? Dimensions of your geometry? What is the information this Figure can deliver? The text from lines 233 to 236 is pure description without interpretation.
17. Line 239 - 250 - the authors simply throw a couple of equations and values without a meaningful derivation of useful quantities. What are the outcomes of the calculation? How does air compare to ice? Be specific!
18. Figure 8 can be significantly simplified - no one cares what a PC and an impedance analyzer look like. Please simply use a block diagram for illustration. In the caption, what do you mean by "possibly" - did you do that or not? Use definitive words in research articles, please.
19. Line 271 - what is "g-paper"?
20. Line 289 - 292 - Please comment on the choices of ice samples (thickness and width) a bit more - how it is related to actual applications in real-world scenarios. Are these sample types representative enough?
21. Figure 9 - you can consider using an error bar instead of a full curve to reflect the standard deviation. The plot is too busy with so many curves. The dash and solid difference are not very obvious. Please try to improve the visibility.
22. Line 316 - 322 - Can you comment on why (ie, physical reasons) the discrimination is more dominant in lower frequency?
23. Line 329 - What is the significance of this coverage factor K? I don't find in the paper describing if this is a figure of merit to look at for this kind of detection. I suggest authors clarify this a bit more.
24. The range of the Y-axis of the capacitance should be identical for all plots such that they can be visually compared by a reader.
25. It is advisable for authors to show a set of sensitivity measurements depending on different quantities of ice on this detector.
26. After reading the paper, I don't see a point in describing the thermal properties of this GNP in section 2.1 as the main focus of the paper should be ice sensing. Authors can consider dropping the section to make the content more focused and compact.
Comments on the Quality of English LanguageIn general good. I have a few comments bulleted above.
Author Response
REV: This paper explored the possibility of using GNP strips for ice detection and removal. I listed a couple of comments below to address –
>AU: We would like to thank the reviewer for useful suggestions and comments, that allowed us to improve the paper.
REV: 1. Figure 12 to Figure 16 are mislabelled as Figure 22 to 66.
>AU: the labels have been corrected
REV: 2. Figure 9 to 16 have very poor resolution. Please fix them.
> AU: these figures have been replaced with better resolution ones.
REV: 3. Line 41 to 50 - references are needed to justify the claims that existing ice sensors "may not capture the full extent..."; "may vary depending on the type..."; "temperature fluctuations..."; "forms of ice..."; "cost". All these are very important metrics to evaluate certain sensors. Therefore, a state-of-the-art literature search has to be comprehensive, such that readers can appreciate why the proposed graphene sensor is good.
> AU: Following the reviewer’s suggestion, the analysis of the state of the art has been improved in the introduction, by better specifying the content of existing references ([3], [5], [7], [9], [15]) and also by adding new references [11]-[14] and [16]-[17]]). The new text is highlighted in yellow in the Introduction (rows 36-63).
REV: 4. Line 63 - typo - "Authors" should be "authors". There should not be a comma before "that".
> AU: Fixed.
REV: 5. Line 76 - typo - "The work" should be "This work".
> AU: Fixed.
REV: 6. Line 82 - typo with an "o" - it seems to be "and"
> AU: Fixed.
REV: 7. Line 118 to 120 - how about an alternative method? Can it achieve similar metrics? Please comment and provide references. Also, does the mentioned parameters (30 um particle, etc) have anything to do with the SEM image in Figure 1? If so, what is the correlation? It is better to be more illustrative.
> AU: This industrial fabrication technique provides GNPs with characteristics similar to those obtained by using other techniques like jet wet milling or microwave irradiation, both suitable for industrial-scale production. We have added a sentence and some references to explain this point (see rows 128–132). In addition, we have clarified the relation between the SEM image and the surface dimensions of the GNP, which can be easily read from the image since a scale bar is provided in it.
REV: 8. Line 136 - What is the physical reason for the negative temperature coefficient? Why it is unusual? Silicon's mobility (lightly doped) also has a negative dependence on temperature. Please comment and provide references for graphene.
> AU: the physical reason for the negative temperature coefficient NTC of the resistance of the graphene strips analyzed in this paper has been briefly explained in the revised manuscript (see rows 150-163). As indicated, when the temperature increases, the mobility within the single flakes increases as a consequence of the increase of the graphene conducting channels, and the mobility between two adjacent flakes increases due to the enhancement of hopping and tunneling mechanics. As suggested, references are given to get more insight on the involved phenomena [45]-[47]. The reviewer is right about the behavior of silicon, but here the term “unusual” refers to conducting materials. Indeed, NTC behavior is easily found in non-conductive or semi-conductive materials (such as silicon or germanium), but is not so usual for conducting materials, as these graphene strips are. A sentence has been added to better highlight this point (see rows 152-154).
REV: 9. Line 137 - why NTC materials are of great interest for sensing? Please justify.
> AU: after the reviewer’s comment, we have found this sentence to be misleading and therefore we have removed it, being inessential to the scope of this paper.
REV: 10. Table 3 - it is advisable to put some other materials's k for comparison such that the benefit of GNP can be shown.
> AU: following this suggestion, and coherently with Table 2, we have provided the thermal parameter values for copper, and we have added a sentence to comment the values (see rows 172-176).
REV. 11. Figure 2 is NOT your original data. I suggest removing it and only citing the values from Ref 40. Otherwise, you should get permission from Ref. 40 to place their data in your paper.
> AU: We have removed the figure and added a new one generated by us by using the data taken from the reference (now Ref.49)
REV. 12. Figure 3 - in the caption, the area 1/2/3 should be specified with words.
> AU: done
REV. 13. Figure 4 - in the zoom-in picture, a couple of labeling of which is which corresponds to Figure 3 would be helpful.
> AU: done
REV. 14. Figure 6 - what are 1,2,3,4,5? Please specify them in your caption so that readers can understand them more easily.
> AU: done
REV. 15. Line 229-232 - No need to show equations (2)-(3) as they are textbook context, nothing new.
> AU: We have removed the equations.
REV. 16. Figure 7 is very poorly presented - What am I looking at? What is the scale bar of the magnitude? Dimensions of your geometry? What is the information this Figure can deliver? The text from lines 233 to 236 is pure description without interpretation.
> AU: We have replaced Fig. 7 with a new figure, adding the scale bars for geometry and for the quantity displayed here. The figure is intended to show the effect of the increase in the dielectric constant when the air is replaced by ice. Specifically, it clearly shows the spread of the electric field lines externally to the sensor, giving the theoretical justification for the observed change in the capacitance. A sentence has been added to explain this concept (see rows 276-283).
REV. 17. Line 239 - 250 - the authors simply throw a couple of equations and values without a meaningful derivation of useful quantities. What are the outcomes of the calculation? How does air compare to ice? Be specific!
> AU: these equations were added for the only purpose to describe the steps followed to extract the capacitance from the solution of the electrostatic problem. Anyway, since this is a standard procedure that can be easily found in textbook, we decided to remove the whole paragraph, leaving a simple sentence (see row 282)
REV. 18. Figure 8 can be significantly simplified - no one cares what a PC and an impedance analyzer look like. Please simply use a block diagram for illustration. In the caption, what do you mean by "possibly" - did you do that or not? Use definitive words in research articles, please.
> AU: We have removed the picture of the instruments alone and added a picture with the whole setup and a particular of the sensor with ice. In addition, we have removed the word “possibly” from the caption.
REV. 19. Line 271 - what is "g-paper"?
> AU: this was a mistake that has been now corrected in “graphene strip”
REV. 20. Line 289 - 292 - Please comment on the choices of ice samples (thickness and width) a bit more - how it is related to actual applications in real-world scenarios. Are these sample types representative enough?
> AU: A sentence has been added (see rows 344-350) to better explain the rationale for the choice of the ice samples, which comes from the requirements of an aeronautical application. Indeed, this work has been developed within an EU R&D project ruled by Airbus (project G-ICE cited in the acknowledgements); therefore, these dimensions are related to cases of interest in the aeronautical field.
REV. 21. Figure 9 - you can consider using an error bar instead of a full curve to reflect the standard deviation. The plot is too busy with so many curves. The dash and solid difference are not very obvious. Please try to improve the visibility
> AU: We have followed this suggestion and have replotted Figs.9-10-11 with error bars representing the standard deviations.
REV. 22. Line 316 - 322 - Can you comment on why (ie, physical reasons) the discrimination is more dominant in lower frequency?
> AU: the discrimination is more dominant in the lower frequency range because, in that range, the difference between the dielectric constants of ice and air is more pronounced, hence the effect of the fringe fields shown in Fig.7b is higher. We have added a sentence (see rows 391-394) to better clarify this point.
REV. 23. Line 329 - What is the significance of this coverage factor K? I don't find in the paper describing if this is a figure of merit to look at for this kind of detection. I suggest authors clarify this a bit more.
> AU: We have added a sentence to explain the meaning coverage factor, also including anew reference, [50] (see rows 375-379):
REV. 24. The range of the Y-axis of the capacitance should be identical for all plots such that they can be visually compared by a reader.
> AU: Following this suggestion, we have re-plotted Figs.9-11 and Figs-14-16 using the same y-axis.
REV. 25. It is advisable for authors to show a set of sensitivity measurements depending on different quantities of ice on this dete
>AU: As pointed out in the reply to remark #20, the ice samples have been chosen with a different aim; therefore, they are not suitable to investigate the correlation between sensitivity and thickness. However, this is a very important step in the development of this idea, which was already decided to carry out in future work. Therefore, we have included this point in the conclusion section as future work (see last sentence in Conclusions).
REV: 26. After reading the paper, I don't see a point in describing the thermal properties of this GNP in section 2.1 as the main focus of the paper should be ice sensing. Authors can consider dropping the section to make the content more focused and compact.
> AU: we agree with the reviewer that the thermal properties reported in Table 3 are not strictly necessary to this work, but the reason why they have been listed here is related to the main goal of this work, namely the demonstration of the possible use as elements for an ice-sensor of the same GNP strips that have been proposed as heating element. The comparison given in Table 3 between the graphene material analyzed here and a typical conducting material (copper) widely used for electro-thermal applications like heating is therefore helping to highlight this message.
Reviewer 3 Report
Comments and Suggestions for Authors
Comments to the authors
The manuscript titled “A capacitive ice-sensor based on graphene nano-platelets strips” by Sibilia et al presented an ice sensor based on the electrical response of thin graphene-based strips. This work is interesting, but there are some minor issues that should be addressed before it is published in “Sensors”.
Minor revision
1. Although the GNP strips are optimized for fabrication, authors need to optimize the parameters of GNP for this new study (ice-sensor application).
2. Three types of ice samples were considered in this study. But what is air A, air B in Figure 9-11... The authors should explain these in the text.
3. The Figure numbers 22-66 should be corrected as 12-16.
4. The authors should add a photograph while using the sensor as an ice detector.
5. The authors should perform stability tests of the proposed ice sensor.
6. The studies below related to the graphene-based sensors are recommended to be cited in the introduction section:
DOI 10.1149/1945-7111/ace33a
DOI 10.1109/JSEN.2023.3246380
Author Response
Rev. 1. Although the GNP strips are optimized for fabrication, authors need to optimize the parameters of GNP for this new study (ice-sensor application).
> AU: As pointed out in the Introduction, we have decided to work with industrial graphene. The following sentence has been added to clarify the scope of the paper and the choice of the material (see rows 73-79):
“Starting from these considerations, the main motivation of this paper is to study the possibility of realizing an ice sensor with the same graphene strips that are used for de-icing. This possibility would open the route to the realization of an integrated system capable of sensing and removing ice. With the scope of studying the feasibility of this idea, this paper analyzes industrial-grade graphene, not optimized for the specific purpose of sensing, but for that of heating.”
Rev. 2. Three types of ice samples were considered in this study. But what is air A, air B in Figure 9-11... The authors should explain these in the text.
> AU: Following this suggestion, we have added the following sentence in the text to explain this point (see rows 381-384):
“The types of ice, in relation to Table 4, are indicated by “A”, “B” and “C”. For air, the same nomenclature is used to indicate the correlation between the tests in the presence and absence of ice performed on the same test day.”
Rev. 3. The Figure numbers 22-66 should be corrected as 12-16.
>AU: This problem came from a wrong conversion from doc to pdf, and has been now fixed.
Rev. 4. The authors should add a photograph while using the sensor as an ice detector.
>AU: A photo of the sensor with ice has been added (see Fig.8b).
Rev. 5. The authors should perform stability tests of the proposed ice sensor.
>AU: The stability analysis has been performed by considering repeated tests with varying conditions. This point has been now better highlighted in the following sentences that have been added to the paper (see rows 351-362):
“In order to realize a metrologically robust characterization of the sensor, a stability analysis was performed. Specifically, in order to assess and estimate the ability of the tested sensor to respond more or less consistently to the same stimulus, 30 measurement tests were conducted for each considered working condition. In this way, it was possible to obtain an indication in terms of sensor stability and repeatability.
In particular, since the causes of possible variations in terms of response can be many and varied (instrumentation, environmental perturbations, real variations in the measurand, etc.), the tests were not only repeated for each operating condition in the presence and absence of ice but also considering variations in environmental conditions. As shown in Section 3, for each test, the response at each frequency was given in terms of the average value over the repeated measurements and the corresponding standard deviation, which is therefore representative of the stability of the sensor.”
REV: 6. The studies below related to the graphene-based sensors are recommended to be cited in the introduction section: DOI 10.1149/1945-7111/ace33a, DOI 10.1109/JSEN.2023.3246380
>AU: These references have been added as refs. [24] and [25] and cited in the introduction, as examples of the use of graphene for sensing applications.
Round 2
Reviewer 2 Report
Comments and Suggestions for Authors
Thanks for the effort of addressing my comments. I think the paper is ready for publication in its current form.
Comments on the Quality of English LanguageNone.